# Use of implementation science models, theories, and frameworks in pediatric rehabilitation: Protocol for a scoping review

Bayley Levy[1,2], Dorothy Luong[1,3], Shauna Kingsnorth[1,4], Iveta Lewis[1,3], Gillian King[1,5], Evdokia Anagnostou[3,6], Nadia Lise Tanel[3], Brayden Levillard[7], Gillian Molzon[2], Himanshi Elugoti[8], Mariam Jawad[9], Sarah Munce[1,3,4,5,10]*

1 Holland Bloorview Kids Rehabilitation Hospital, Bloorview Research Institute, Toronto, Ontario, Canada, 2 Central Michigan University College of Medicine, Mount Pleasant, Michigan, United States of America, 3 Holland Bloorview Kids Rehabilitation Hospital, Toronto, Ontario, Canada, 4 Rehabilitation Sciences Institute, University of Toronto, Toronto, Ontario, Canada, 5 Department of Occupational Science and Occupational Therapy, University of Toronto, Toronto, Canada, 6 Department of Pediatrics, Temerty Faculty of Medicine, University of Toronto, Toronto, Ontario, Canada, 7 Queens University, Kingston, Ontario, Canada, 8 University of Waterloo, Waterloo, Ontario, Canada, 9 McMaster University, Hamilton, Ontario, Canada, 10 Institute of Health Policy, Management and Evaluation, University of Toronto, Toronto, Ontario, Canada

* smunce@hollandbloorview.ca

## Abstract

### Introduction

Implementation science frameworks – including process models, determinant frameworks, classic theories, implementation theories, and evaluation frameworks – are increasingly used to guide the translation of evidence-based interventions into practice. In paediatric rehabilitation, where interventions are complex and often require multidisciplinary collaboration, these frameworks can support systematic and context-sensitive implementation. However, the extent to which these frameworks have been used has not been comprehensively reviewed.

### Objective

Determine the extent, nature, and specific contexts of the existing literature on the use of implementation science models, theories, and/or frameworks (MTFs) in paediatric rehabilitation.

### Methods

This scoping review will follow the Joanna Briggs Institute (JBI) methodological guidance for scoping reviews. A comprehensive search strategy will be developed with a health sciences librarian and applied across multiple electronic databases: MEDLINE (Ovid), Embase, CINAHL, PsycINFO, ACM Digital Library, Web of Science, the Cochrane Central Register of Controlled Trials, PEDro, and RehabData. We will

**Data availability statement:** No datasets were generated or analysed during the current study. All relevant data from this study will be made available upon study completion.

**Funding:** This work is supported and funded by Holland Bloorview Kids Rehabilitation Hospital Foundation. Dr. Munce holds a CIHR Implementation Science Chair in Human Development, Child and Youth Health. The funders did not have any role in study design, data collection and analysis, decision to publish, or preparation of the manuscript. There was no additional external funding received for this study.

**Competing interests:** The authors have declared that no competing interests exist.

search English language articles published since 2006. Studies will be included if they report on the application of implementation science MTFs in the context of paediatric rehabilitation. Screening of titles and abstracts and full texts will be performed independently and in duplicate using Covidence. Discrepancies will be resolved through discussion or a third reviewer. Data will be extracted using a standardized form. Quantitative data will be summarized using numerical counts. Qualitative data will be analyzed using content analyses.

## Results

This review will report on the use of implementation science MTFs in paediatric rehabilitation, identifying trends on the specific types applied, highlight gaps and/or underutilization across domains or developmental stages, and potentially uncover emerging frameworks. Finally, the results may inform the development of future implementation strategies and capacity-building initiatives within the field.

## Introduction

Paediatric rehabilitation is a highly specialized and continually evolving field that requires complex diagnostic and therapeutic modalities in treating children and youth with diverse developmental, physical, and cognitive disabilities. The prevalence of childhood disability has steadily increased since the 1990s [1], and as of 2025, approximately 1 in 6 children are living with a disability requiring short and/or long-term services [2]. Paediatric rehabilitation interventions are inherently complex due to the interplay of multiple intersecting factors across the individual, organizational, and community levels [3]. These interventions involve coordination across multi- and inter-disciplinary healthcare teams, providing care across dynamic settings such as the home, clinic, hospital, school, and within the community [2–4]. Furthermore, these interventions have varying targets/aims, including but not limited to social support, education, improving functioning, increasing participation, promoting patient autonomy, among many others. However, a persistent issue in paediatric rehabilitation services is the fragmentation of care; although children may be under the care of multidisciplinary teams, treatment plans lack a "whole-child perspective" [4], which are limited in the integration of shared decision-making and patient-specific interventions. Consequently, disjointed care may result in poorer health outcomes for paediatric rehabilitation patients, such as missed opportunities in development, reduced engagement in services, increased caregiver burden, and negative impacts on the quality-of-life of the paediatric patient and family/caregivers.

Implementation science remains underutilized in the field of paediatric rehabilitation despite its potential, for example, improving the uptake of evidence-based interventions using a systematic, coordinated, and context-sensitive approach [3,4]. More specifically, the application of implementation science models, theories, and/or frameworks (MTFs) can help to bridge the gap between evidence-based interventions and effective mobilization into clinical practice [2,5]. Implementation science MTFs can

help identify barriers and facilitators to implementation at various levels (e.g., patients, clinicians, organization), evaluate not only what works, but also how and why it works (or does not), and help identify what factors influence sustainability (e.g., leadership support, workflow fit).

Implementation science MTFs include five categories of theoretical models: process models, determinant frameworks, classic theories, implementation theories, and evaluation frameworks, which are based on the taxonomy proposed by Nilsen (2015) [5]. Process models outline sequential steps for translating research into practice, such as the Knowledge-to-Action (KTA) Framework [6] and the Knowledge Model of Knowledge Translation [7]. Determinant frameworks identify multilevel barriers and facilitators that influence implementation outcomes, though they often lack causal explanations [8,9]. Classic theories, drawn from disciplines like psychology and sociology, describe mechanisms of change but are considered "passive" as they do not directly guide implementation efforts [5]. In contrast, implementation theories, such as the Normalization Process Theory [10], offer specific insights into the integration of new practices. Finally, evaluation frameworks, such as RE-AIM (Reach, Effectiveness, Adoption, Implementation, and Maintenance) provide structured approaches for assessing implementation outcomes [5] and are useful in assessing implementation success and providing outcome categories/criteria for evaluation.

In a recent scoping review, Ghahramani et al. in 2025 [2] summarized the characteristics, implementation strategies, and outcomes of implementation studies in paediatric rehabilitation. Their findings highlighted the significant underutilization of implementation strategies, noting that nearly half of the 44 included studies used the implementation strategy, "train and educate key informant" while 19 studies adopted "use evaluative/iterative strategies". However, the review did not explore a broader application of implementation science MTFs in paediatric rehabilitation.

Indeed, across the existing literature, findings highlight the need for more structured, system-level approaches that leverage implementation science frameworks to support effective knowledge mobilization in paediatric rehabilitation [11–13]. Educational strategies remain the most frequently employed, as previously shown, and while they can enhance provider knowledge, skills, and attitudes, they often fall short in facilitating the broader system-level changes required to support the uptake of emerging interventions [2]. Lazarowitz et al. (2025) demonstrated the inconsistent reporting and limited use of MTFs in paediatric rehabilitation, while emphasizing the importance of multifaceted strategies in supporting successful uptake of interventions [11]. Anaby et al. (2021) highlighted the critical role of knowledge user collaboration in their study examining factors in accelerating the uptake of interventions into clinical paediatric rehabilitation practice [12]. Gefen et al. (2024) also described the limited reporting and underutilization of implementation science MTFs, as previously stated, while also bringing attention the need for more system-level approaches, and the coordination of diverse knowledge users to support successful implementation strategies [13].

Despite the recognized importance of implementation science frameworks in translating evidence into practice, these MTFs are not being leveraged to their full potential in the field of paediatric rehabilitation [14]. Without a clear understanding of how and which implementation science MTFs are applied, opportunities to improve the uptake of clinical interventions and sustainability for long-term goals may be missed. Thus, the objective of this scoping review is to determine the extent, nature, and specific contexts of the literature on the use of implementation science MTFs within paediatric rehabilitation.

## Methods

### Design

The scoping review will be conducted in accordance with the Joanna Briggs Institute (JBI) methodology for scoping reviews. The development of this scoping review protocol follows the JBI "best practice guidance and reporting items for the development of scoping review protocols" [15]. Reporting will follow the Preferred Reporting Items for Systematic Reviews and Meta-Analyses extension for Scoping Reviews (PRISMA-ScR) [16,17]. The protocol has been registered with Open Science Framework (osf.io/fn9ku) [18].

This scoping review will capture and align with the core elements of the Population, Concept, and Context (PCC) framework of inclusion criteria [16]. The population of interest involves paediatric patients (age 18 years of age and younger) with physical and cognitive disabilities, including but not limited to: congenital brain injuries, cerebral palsy, spina bifida, attention-deficit hyperactivity disorder, autism spectrum disorder, communication disorders, intellectual and motor disabilities, and learning disabilities. Additionally, studies involving the families of paediatric patients with disabilities and healthcare professionals/providers working within paediatric rehabilitation are of interest. The context of interest is paediatric rehabilitation. Finally, the concept of this scoping review is the use of implementation science MTFs.

## Search strategy

A comprehensive literature search has been developed in collaboration with a medical librarian at Holland Bloorview Kids Rehabilitation Hospital (see search strategy in S1 Appendix A: Ovid Search in S1 File). The final search strategy also underwent peer review using the Peer Review of Electronic Search Strategies (PRESS) statement checklist [19] with another librarian. The research team engaged in consensus meetings with the medical librarian to refine the search strategy. The search terms have been carefully selected and are consistent with previous reviews conducted on this topic [2,11–13]. The search strategy will be adapted for each database utilized. A preliminary search for existing scoping reviews was conducted on the JBI Database of Systematic Reviews and Implementation Reports, Episteminokis Databases, Open Science Framework and PROSPERO in May 2025. No reviews were published or currently underway on the use of implementation science MTFs within paediatric rehabilitation.

A two-step search strategy will be utilized in this review. An initial limited search of two interfaces (1) Ovid MEDLINE and (2) CINAHL (EBSCO) was undertaken to identify articles on the topic and as a search validation procedure to ensure the comprehensiveness and reliability of our approach. The initial search strategy was developed and tested in MEDLINE using controlled vocabulary (e.g., MeSH terms) to identify additional key search terms commonly found within the titles, abstracts, and full texts of similar articles. Key search terms will include variations of *implementation science*, specific models, theories, and frameworks (e.g., CFIR, EPIS, Theoretical Domains Framework), and relevant conditions and populations (e.g., "paediatric rehabilitation", "cerebral palsy", "autism spectrum disorder", "developmental disabilities", "rehabilitation", "therapy", "brain injury", "neurodevelopmental disorders"). This strategy was developed to ensure all relevant studies on the application of implementation science MTFs in paediatric rehabilitation are captured.

Database searches will be conducted across: MEDLINE (Ovid), Embase, CINAHL, PsycINFO, ACM Digital Library, Web of Science, the Cochrane Central Register of Controlled Trials, PEDro, and RehabData. In addition to the primary database search, the research team will conduct forward citation searching (descendancy) to identify articles that have been cited by included studies and backward citation searching (ascendancy) to screen the reference lists of included studies to ensure all relevant sources have been captured. In cases where full texts are not available or accessible, contact authors will be e-mailed to attempt to source the articles. Communication will be attempted 2–3 times, and these efforts will be reported in the final manuscript with their associated outcomes.

## Evidence screening and selection

All articles retrieved from the database search will be imported into Covidence [20] and duplicates will be removed.

English studies published since 2006 will be eligible for inclusion. The year 2006 was chosen as it corresponds with the release of the *Implementation Science* journal, a seminal journal in the field. Studies must report on the application of an implementation science MTF in the context of paediatric rehabilitation (including studies involving the families of paediatric patients with disabilities and healthcare professionals/providers working within this field) will be included. Within the field of implementation science and elsewhere, "framework", "model", and "theory" are often used interchangeably; however, Nilsen (2015) provided a taxonomy to conceptualize and provide clarity amongst these structurally similar terms that have nuanced variations [5,6], as previously described. Specifically, theory refers to "a set of analytical principles or statements

designed to structure our observation, understanding, and explanation of the world," typically characterized by defined constructs, relationships between variables, and predicted outcomes; model refers to "a deliberate simplification of a phenomenon," which is descriptive but does not explain causal relationships; and framework is defined as "a structure, overview, outline, system, or plan consisting of various descriptive categories and the relations between them," offering a categorization of phenomena without explanatory mechanisms. For the purposes of this scoping review, we will adhere to these definitions to ensure conceptual clarity and consistency. Additionally, we have designed an inclusive search strategy that accounts for the overlap of these terms. Specifically, we have incorporated the most commonly used implementation science MTFs by name. This approach aims to optimize both the sensitivity and specificity of our search and ensure that relevant literature is accurately captured and appropriately classified. All primary study designs are eligible for inclusion, while non-primary designs (e.g., editorials, opinion pieces, commentaries, protocols, systematic reviews, scoping reviews, and meta-analyses) will be excluded.

Level 1 (i.e., title and abstract) and level 2 (i.e., full text) screening will be completed in duplicate by two independent reviewers using the eligibility criteria. Prior to starting level 1 screening, the inclusion criteria will be piloted on a random sample of 10% of citations. The pilot test will be performed to confirm there is consistent interpretation and application of the inclusion and exclusion criteria. The eligibility of the defined inclusion criteria may be revised to improve the consistency, if deemed necessary by the research team, or if a low inter-rater agreement (i.e., below 70%) is observed. Similarly, a pilot test of 10% will be done prior to full level 2 screening begins. At each stage, discrepancies will be resolved by a discussion between reviewers, and in instances in which a consensus cannot be reached, a third reviewer will adjudicate.

## Data extraction

Data extraction will be conducted in duplicate by two independent reviewers using the data extraction tool on Covidence. A pilot test of the full data extraction will be completed on approximately 10 articles chosen at random to ensure reliability and accuracy among reviewers. The pilot test will identify any discrepancies in the data extraction process and provide an opportunity to modify the template to enhance inter-rater consistency. Any disagreements that arise between the reviewers will be resolved through discussion, or with an additional reviewer. If appropriate, authors of papers will be contacted to request missing or additional data, where required.

Data extraction will follow an iterative approach, with the final categories being defined progressively as the authors gain a deeper understanding of the data. It is anticipated that categories will include: implementation science MTF used, type of implementation science MTF (based on Nilsen (2015)'s taxonomy) [5,6], type of paediatric rehabilitation, disability population, and study findings including implementation outcomes, as well as other study characteristics (e.g., author, year of publication, name of journal, country, study design, setting/care context, population, sample size, objective) and additional population characteristics, including PROGRESS-Plus characteristics (Cochrane Methods Equity) [21,22] (e.g., place of residence, race/ethnicity/language, occupation status, gender/sex, religion, education, socioeconomic status, social capital, personal characteristics associated with discrimination, features of relationships, time dependent relationships).

## Data analysis and presentation

This scoping review will follow the 2022 JBI methodological guidance [15] to ensure best practices in data analysis and presentation. Quality and risk of bias will not be assessed. Findings will be interpreted with consideration of study design and methodological characteristics, with emphasis on mapping the extent and nature of the evidence. Descriptive numerical summaries will be conducted to present characteristics of included studies by frequency and proportion (e.g., number of studies, year of publication, and types of implementation science MTFs used). Content analysis will be used to synthesize qualitative data within the included studies [23]. This analysis will focus on: (1) types of MTFs employed (based on Nilsen's taxonomy [5,6]; (2) how the constructs in the MTF were operationalized; (3) characteristics of target populations

(e.g., children with disabilities, paediatric healthcare professionals); (4) the intervention focus (if applicable); (5) the care setting; and (6) outcomes reported (if applicable). Equity-related variables will also be analyzed descriptively using the PROGRESS-Plus framework where applicable, with outputs including summaries of reported variables and frequencies across studies. This will help identify how social determinants such as place of residence, gender, or socioeconomic factors influence the success of the implementation of the paediatric rehabilitation interventions. These insights may reveal disparities in how MTFs are applied across different contexts [19,20]. Findings from both the quantitative summaries and content analysis will be presented narratively and visually, using tables, framework maps, and charts, to clearly convey observed patterns, trends, and gaps in the literature.

Level 1 screening is currently underway and expected to be completed by the end of October 2025; with Level 2 screening completion by December 2025. Data extraction is expected to immediately follow the completion of Level 2 screening and be completed by March 2026. Data analysis and manuscript writing will occur concurrently, and we expect to produce the final manuscript by June 2026.

## Discussion and Conclusion

This scoping review will provide a comprehensive overview of the extent, nature, and specific contexts of the literature on the use of implementation science MTFs in paediatric rehabilitation. By mapping the current evidence, this review will identify which MTFs are most commonly applied, how they are operationalized, and where gaps exist in their application, adaptation, or development. For consistency, we will follow the definitions of MTFs outlined by Nilson (2015), where a theory is an explanatory set of principles with specified relationships and predicted outcomes; a model is a descriptive representation of a phenomenon; and a framework is a structured overview that categorizes concepts without specifying causal mechanisms [5,6].

To disseminate our findings, we will use a variety of passive and active end-of-grant knowledge mobilization approaches. Traditional knowledge mobilization will include dissemination through meetings locally, nationally, and internationally (e.g., European Implementation Conference, Annual Conference on the Science of Dissemination and Implementation in Health) and publications in a peer-reviewed journal (e.g., Implementation Science Research and Practice). Furthermore, findings from this scoping review will be used to inform an implementation science and practice pathway at Holland Bloorview Kids Rehabilitation Hospital, which in turn could be used at other paediatric rehabilitation institutions.

A strength of this study is the broad inclusion criteria, allowing for a comprehensive map of the current literature on implementation science in paediatric rehabilitation. The search strategy had undergone peer review using the PRESS statement checklist, which further strengthens the relevance, comprehensiveness, and quality of the search strategy. Furthermore, screening and extraction were conducted in duplicate, ensuring data reliability. A limitation of this study is the inclusion of articles published in English, thereby potentially excluding relevant studies published in other languages. We also acknowledge that the proposed timeline may be ambitious given the scope of the review. To mitigate potential delays, we have assembled a large team of experienced reviewers with prior expertise in conducting scoping reviews of similar scope. In addition, we will implement structured workflows, including regular team check-ins, and predefined milestones to closely monitor progress and address any delays early. Another limitation of this study is that this review focuses on studies that explicitly reference implementation science MTFs, thus it may not capture broader implementation efforts that do not formally apply these frameworks.

This scoping review findings will inform researchers, clinicians, and decision-makers about the state of implementation science in paediatric rehabilitation and guide more effective, theory-informed implementation efforts. Ultimately, this review aims to strengthen the use of implementation science in advancing the uptake of evidence-based practices, improving care delivery, and addressing the unique needs of children and families in paediatric rehabilitation settings.

## Supporting information

**S1 File. Appendix A: Ovid Search Strategy.** Search strategy for OVID-MEDLINE Database.
(DOCX)

**S2 File. Preferred Reporting Items for Systematic Review and Meta-Analysis Protocols (PRISMA-P) checklist.**
PRISMA-P checklist completed for scoping review protocol.
(DOCX)

## Acknowledgments

We would like to thank Jessie Cunningham, a librarian at the Hospital for Sick Children, for their support in completing the PRESS Peer Review of Electronic Search Strategies for the search strategy developed for this review.

## Author contributions

**Conceptualization:** Sarah Munce.

**Data curation:** Iveta Lewis.

**Funding acquisition:** Sarah Munce.

**Investigation:** Bayley Levy, Brayden Levillard, Gillian Molzon, Himanshi Elugoti, Mariam Jawad, Sarah Munce.

**Methodology:** Sarah Munce.

**Project administration:** Bayley Levy, Dorothy Luong, Sarah Munce.

**Resources:** Bayley Levy, Dorothy Luong, Iveta Lewis, Sarah Munce.

**Software:** Iveta Lewis.

**Supervision:** Bayley Levy, Dorothy Luong, Sarah Munce.

**Validation:** Iveta Lewis.

**Writing – original draft:** Bayley Levy.

**Writing – review & editing:** Bayley Levy, Dorothy Luong, Shauna Kingsnorth, Iveta Lewis, Gillian King, Evdokia Anagnostou, Nadia Lise Tanel, Brayden Levillard, Gillian Molzon, Himanshi Elugoti, Mariam Jawad, Sarah Munce.

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
