## [Decision Letter · Decision Letter 0]

9 Feb 2026

PONE-D-25-39248Use of Implementation Science Models, Theories, and Frameworks in Pediatric Rehabilitation: Protocol for a Scoping ReviewPLOS One

Dear Dr. Munce,

Thank you for submitting your manuscript to PLOS ONE. After careful consideration, we feel that it has merit but does not fully meet PLOS ONE’s publication criteria as it currently stands. Therefore, we invite you to submit a revised version of the manuscript that addresses the points raised during the review process.

Protocol is well developed, methodologically sound, and aligned with established scoping review standards.

1. You include a broad range of MTFs and pediatric populations, which enhances inclusivity but may also introduce conceptual heterogeneity. Consider clarifying how the team will manage conceptual overlap and variability across frameworks during synthesis to ensure interpretive coherence.

2. You state that studies must report “application” of MTFs, but the criteria for determining meaningful use versus superficial citation are not fully specified. clarify how the reviewers will distinguish substantive application from nominal or symbolic reference to frameworks.

3. While deductive content analysis is described, details regarding coding procedures, analyst reflexivity, and consensus-building are limited.

4. You exclude formal quality appraisal, consistent with scoping review standards. However, the implications of this decision are not fully discussed. Consider briefly outlining how variation in study rigor will be considered during interpretation.

5.  Your proposed timeline is a little ambitious given the scope of databases and analytic plans. While its acceptable, a brief acknowledgment of potential timeline risks and mitigation strategies may enhance credibility.

6. Terminology related to “model,” “theory,” and “framework” could be briefly reiterated in the

7. The description of equity analysis could be slightly expanded to clarify anticipated outputs.

We look forward to receiving your revised manuscript.

Kind regards,

Udoka Okpalauwaekwe, MD, MPH, PhD

Academic Editor

PLOS One

Journal Requirements:

1. When submitting your revision, we need you to address these additional requirements. Please ensure that your manuscript meets PLOS ONE's style requirements, including those for file naming. The PLOS ONE style templates can be found at https://journals.plos.org/plosone/s/file?id=wjVg/PLOSOne_formatting_sample_main_body.pdf and https://journals.plos.org/plosone/s/file?id=ba62/PLOSOne_formatting_sample_title_authors_affiliations.pdf 2. Thank you for stating in your Funding Statement: This work is supported and funded by Holland Bloorview Kids Rehabilitation Hospital Foundation. Dr. S. Munce holds a CIHR Implementation Science Chair in Human Development, Child and Youth Health. The funders did not have any role in study design, data collection and analysis, decision to publish, or preparation of the manuscript.  Please provide an amended statement that declares *all* the funding or sources of support (whether external or internal to your organization) received during this study, as detailed online in our guide for authors at ://journals.plos.org/plosone/s/submit-now. Please also include the statement “There was no additional external funding received for this study.” in your updated Funding Statement. Please include your amended Funding Statement within your cover letter. We will change the online submission form on your behalf. 3. Thank you for stating the following in the Acknowledgments Section of your manuscript: We would like to thank, Jessie Cunningham, a librarian at the Hospital for Sick Children, for their support in completing the PRESS Peer Review of Electronic Search Strategies for the search strategy developed for this review. The authors would also like to thank the Holland Bloorview Kids Rehabilitation Hospital Foundation for supporting this work. We note that you have provided funding information that is not currently declared in your Funding Statement. However, funding information should not appear in the Acknowledgments section or other areas of your manuscript. We will only publish funding information present in the Funding Statement section of the online submission form. Please remove any funding-related text from the manuscript and let us know how you would like to update your Funding Statement. Currently, your Funding Statement reads as follows: This work is supported and funded by Holland Bloorview Kids Rehabilitation Hospital Foundation. Dr. S. Munce holds a CIHR Implementation Science Chair in Human Development, Child and Youth Health. The funders did not have any role in study design, data collection and analysis, decision to publish, or preparation of the manuscript. Please include your amended statements within your cover letter; we will change the online submission form on your behalf. 4. When completing the data availability statement of the submission form, you indicated that you will make your data available on acceptance. We strongly recommend all authors decide on a data sharing plan before acceptance, as the process can be lengthy and hold up publication timelines. Please note that, though access restrictions are acceptable now, your entire data will need to be made freely accessible if your manuscript is accepted for publication. This policy applies to all data except where public deposition would breach compliance with the protocol approved by your research ethics board. If you are unable to adhere to our open data policy, please kindly revise your statement to explain your reasoning and we will seek the editor's input on an exemption. Please be assured that, once you have provided your new statement, the assessment of your exemption will not hold up the peer review process. 5. We notice that your supplementary table is included in the manuscript file. Please remove, leaving only the individual file, uploaded separately. Please ensure that each Supporting Information file has a legend listed in the manuscript after the references list. 6. If the reviewer comments include a recommendation to cite specific previously published works, please review and evaluate these publications to determine whether they are relevant and should be cited. There is no requirement to cite these works unless the editor has indicated otherwise.

Additional Editor Comments:

Protocol is well developed, methodologically sound, and aligned with established scoping review standards.

1. You include a broad range of MTFs and pediatric populations, which enhances inclusivity but may also introduce conceptual heterogeneity. Consider clarifying how the team will manage conceptual overlap and variability across frameworks during synthesis to ensure interpretive coherence.

2. You state that studies must report “application” of MTFs, but the criteria for determining meaningful use versus superficial citation are not fully specified. clarify how the reviewers will distinguish substantive application from nominal or symbolic reference to frameworks.

3. While deductive content analysis is described, details regarding coding procedures, analyst reflexivity, and consensus-building are limited.

4. You exclude formal quality appraisal, consistent with scoping review standards. However, the implications of this decision are not fully discussed. Consider briefly outlining how variation in study rigor will be considered during interpretation.

5. Your proposed timeline is a little ambitious given the scope of databases and analytic plans. While its acceptable, a brief acknowledgment of potential timeline risks and mitigation strategies may enhance credibility.

6. Terminology related to “model,” “theory,” and “framework” could be briefly reiterated in the

7. The description of equity analysis could be slightly expanded to clarify anticipated outputs.

Reviewers' comments:

Reviewer's Responses to Questions

Comments to the Author

1. Does the manuscript provide a valid rationale for the proposed study, with clearly identified and justified research questions?

Reviewer #1: Yes

2. Is the protocol technically sound and planned in a manner that will lead to a meaningful outcome and allow testing the stated hypotheses?

Reviewer #1: Yes

3. Is the methodology feasible and described in sufficient detail to allow the work to be replicable?

Reviewer #1: Yes

4. Have the authors described where all data underlying the findings will be made available when the study is complete?

Reviewer #1: Yes

5. Is the manuscript presented in an intelligible fashion and written in standard English?

Reviewer #1: Yes

6. Review Comments to the Author

You may also provide optional suggestions and comments to authors that they might find helpful in planning their study.

Reviewer #1: Great protocol and very well written. My recommendations are:

- Adding the rationale for why a certain MTF was chosen in a paper and the fidelity of implementation of the MTF as a category for the data to be abstracted

- Consideration for analyzing the data to see if there are difference between low/medium income country vs. high income country

- Fleshing out the discussion and conclusion section - ex. adding details about how the results would be disseminated, strengths and limitations. It is currently only a paragraph

- A thought that I had when reading this protocol that it may become a case of "we don't know what we don't know". I don't know whether it is feasible to determine the extent of the literature on the use of MTFs. For example, what if someone wrote a paper detailing how they implemented a model of care, but didn't use implementation science MTF at all? I am not clear that the search strategy would pick it up.

7. PLOS authors have the option to publish the peer review history of their article (what does this mean?). If published, this will include your full peer review and any attached files.

Do you want your identity to be public for this peer review? For information about this choice, including consent withdrawal, please see our Privacy Policy.

Reviewer #1: No

To ensure your figures meet our technical requirements, please review our figure guidelines: s://journals.plos.org/plosone/s/figures

You may also use PLOS’s free figure tool, NAAS, to help you prepare publication quality figures: s://journals.plos.org/plosone/s/figures#loc-tools-for-figure-preparation.

---

## [Author Response · Author response to Decision Letter 1]

27 Mar 2026

We would like to thank the reviewer and editor for their feedback on our manuscript. Here are our responses to the feedback (and attached as a document in this resubmission).

Journal Requirements:

Response: Thank you for your comment. We have reformatted our manuscript to meet PLOS ONE’s style requirements including file naming.

This work is supported and funded by Holland Bloorview Kids Rehabilitation Hospital Foundation. Dr. S. Munce holds a CIHR Implementation Science Chair in Human Development, Child and Youth Health. The funders did not have any role in study design, data collection and analysis, decision to publish, or preparation of the manuscript.

Response: We have updated our Funding Statement in our cover letter and below:

“This work is supported and funded by Holland Bloorview Kids Rehabilitation Hospital Foundation. Dr. Munce holds a CIHR Implementation Science Chair in Human Development, Child and Youth Health. The funders did not have any role in study design, data collection and analysis, decision to publish, or preparation of the manuscript. There was no additional external funding received for this study.”

We would like to thank, Jessie Cunningham, a librarian at the Hospital for Sick Children, for their support in completing the PRESS Peer Review of Electronic Search Strategies for the search strategy developed for this review. The authors would also like to thank the Holland Bloorview Kids Rehabilitation Hospital Foundation for supporting this work.

This work is supported and funded by Holland Bloorview Kids Rehabilitation Hospital Foundation. Dr. S. Munce holds a CIHR Implementation Science Chair in Human Development, Child and Youth Health. The funders did not have any role in study design, data collection and analysis, decision to publish, or preparation of the manuscript.

Response: We have revised our Funding Statement based on the above comment and removed any funding-related text from the manuscript. The updated Funding Statement has been included in our cover letter.

Response: This manuscript is a protocol paper, thus, no datasets were generated or analyzed. Relevant data from this study will be made available in a separate manuscript. No changes have been made to the data availability statement.

5. We notice that your supplementary table is included in the manuscript file. Please remove, leaving only the individual file, uploaded separately. Please ensure that each Supporting Information file has a legend listed in the manuscript after the references list.

Response: Thank you. We have removed S1 Appendix A from the main manuscript and have uploaded it separately.

Response: Thank you.

Response: Thank you. We have updated the reference list to include the additional relevant citation and have incorporated it into the manuscript accordingly.

Additional Editor Comments:

Protocol is well developed, methodologically sound, and aligned with established scoping review standards.

1. You include a broad range of MTFs and pediatric populations, which enhances inclusivity but may also introduce conceptual heterogeneity. Consider clarifying how the team will manage conceptual overlap and variability across frameworks during synthesis to ensure interpretive coherence.

Response: Thank you for your comments. As this scoping review aims to map the use of implementation science MTFs in pediatric rehabilitation, we will assess heterogeneity during data synthesis, including variation in both MTFs and pediatric populations. To support consistency in categorization of the MTFs, we will use Nilsen (2015)’s taxonomy to group MTFs into the following categories: process models, determinant frameworks, classic theories, implementation theories, and evaluation frameworks. MTFs that do not belong to any of these categories will be put into the “other MTFs” category when applicable. We expect limited range of implementation science MTFs to be used as well as higher usage of common implementation science MTFs (e.g., CFIR, RE-AIM), so we expect that Nilsen (2015)’s taxonomy to categorize the MTFs will be very useful.

To manage conceptual overlap and variability, we will chart how the implementation science MTFs were applied across studies and describe similarities and differences within and across categories. This approach will allow us to provide a structured overview of MTF use while being consistent with scoping review methodology.

We provide details related to Nilsen’s taxonomy on pages 4-5, lines 86-99:

“Implementation science MTFs include five categories of theoretical models: process models, determinant frameworks, classic theories, implementation theories, and evaluation frameworks, which are based on the taxonomy proposed by per Nilsen (2015) [5]. Process models outline sequential steps for translating research into practice, such as the Knowledge-to-Action (KTA) Framework [6] and the Knowledge Model of Knowledge Translation [7]. Determinant frameworks identify multilevel barriers and facilitators that influence implementation outcomes, though they often lack causal explanations [8,9]. Classic theories, drawn from disciplines like psychology and sociology, describe mechanisms of change but are considered “passive” as they do not directly guide implementation efforts [5]. In contrast, implementation theories, such as the Normalization Process Theory [10], offer specific insights into the integration of new practices. Finally, evaluation frameworks, such as RE-AIM (Reach, Effectiveness, Adoption, Implementation, and Maintenance) provide structured approaches for assessing implementation outcomes [5] and are useful in assessing implementation success and providing outcome categories/criteria for evaluation.”

We have provided additional details on how we will categorize the implementation science MTFs at data extraction on page 11, lines 236-245:

“It is anticipated that categories will include: implementation science MTF used, type of implementation science MTF (based on Nilsen (2015)’s taxonomy) [5, 6], type of pediatric rehabilitation, disability population, and study findings including implementation outcomes, as well as other study characteristics (e.g., author, year of publication, name of journal, country, study design, setting/care context, population, sample size, objective) and additional population characteristics, including PROGRESS-Plus characteristics (Cochrane Methods Equity) [19, 20] (e.g., place of residence, race/ethnicity/language, occupation status, gender/sex, religion, education, socioeconomic status, social capital, personal characteristics associated with discrimination, features of relationships, time dependent relationships).”

2. You state that studies must report “application” of MTFs, but the criteria for determining meaningful use versus superficial citation are not fully specified. clarify how the reviewers will distinguish substantive application from nominal or symbolic reference to frameworks.

Response: Our objective is to understand what the current extent of the literature is with respect to the use of implementation MTFs. As such, we decided to not restrict inclusion based on level of “meaningful” application of the MTF. Studies will be included if they report any use of an implementation MTF, which will allow us to capture the range of reported uses. During data analysis we will be able to report on whether research teams have been “meaningfully” applying implementation MTFs in their study.

3. While deductive content analysis is described, details regarding coding procedures, analyst reflexivity, and consensus-building are limited.

Response: We meant to say that we will conduct content analyses instead of deductive content analyses. As previously mentioned, we will use Nilsen (2015)’s taxonomy to categorize the implementation science MTFs and conduct content analysis to synthesize qualitative data of the included studies.

We have made these changes on page 11-12, lines 251-254:

“Descriptive numerical summaries will be conducted to present characteristics of included studies by frequency and proportion (e.g., number of studies, year of publication, and types of implementation science MTFs used). Content analysis will be used to synthesize qualitative data within the included studies [23].”

4. You exclude formal quality appraisal, consistent with scoping review standards. However, the implications of this decision are not fully discussed. Consider briefly outlining how variation in study rigor will be considered during interpretation.

Response: While formal quality appraisal is not required in scoping reviews, we agree that variation in study rigor should be considered during the data synthesis and interpretation. We have revised the manuscript to clarify that findings will be interpreted in light of study design and methodological characteristics (e.g., study type, sample size, and reporting detail) on page 11, lines 249-251. Our conclusions will emphasize the mapping of evidence and identification of patterns and gaps, rather than making judgments about the strength of evidence. We will also discuss areas where the current evidence base is limited or heterogeneous.

“ Findings will be interpreted with consideration of study design and methodological characteristics, with emphasis on mapping the extent and nature of the evidence.

5. Your proposed timeline is a little ambitious given the scope of databases and analytic plans. While its acceptable, a brief acknowledgment of potential timeline risks and mitigation strategies may enhance credibility.

Response: We acknowledge that the proposed timeline is ambitious given the scope of databases and analytic approach. To mitigate potential delays, we have assembled a large team of experienced reviewers with prior expertise in conducting scoping reviews of similar scope. In addition, we will implement structured workflows, including regular team check-ins, and predefined milestones to closely monitor progress and address any delays early.

We have included this in our manuscript on page 14, lines 302-305

“To mitigate potential delays, we have assembled a large team of experienced reviewers with prior expertise in conducting scoping reviews of similar scope. In addition, we will implement structured workflows, including regular team check-ins, and predefined milestones to closely monitor progress and address any delays early.”

6. Terminology related to “model,” “theory,” and “framework” could be briefly reiterated in the

Response: We have briefly restated the definitions of “model,” “theory,” and “framework” in our discussion section on page 13, lines 277-284:

“This scoping review will provide a comprehensive overview of the extent, nature, and specific contexts of the literature on the use of implementation science MTFs in pediatric rehabilitation. By mapping the current evidence, this review will identify which MTFs are most commonly applied, how they are operationalized, and where gaps exist in their application, adaptation, or development. For consistency, we will follow the definitions of MTFs outlined by Nielson (2015), where a theory is an explanatory set of principles with specified relationships and predicted outcomes; a model is a descriptive representation of a phenomenon; and a framework as a structured overview that categorizes concepts without specifying causal mechanisms [5,6].”

7. The description of equity analysis could be slightly expanded to clarify anticipated outputs.

Response: Thank you for this suggestion. We have expanded the description of the equity analysis to clarify anticipated outputs. Specifically, we now describe how equity-related variables will be summarized descriptively using the PROGRESS-Plus framework and its implications within paediatric rehabilitation on page 12, lines 258-263:

“Equity-related variables will also be analyzed descriptively using the PROGRESS-Plus framework where applicable, with outputs including summaries of reported variables and frequencies across studies. This will help identify how social determinants such as place of residence, gender, or socioeconomic factors influence the success of the implementation of the paediatric rehabilitation interventions.”

Reviewers' comments:

Reviewer #1: Great protocol and very well written.

My recommendations are:

Adding the rationale for why a certain MTF was chosen in a paper and the fidelity of implementation of the MTF as a category for the data to be abstracted

Response: Thank you for this comment! We will be sure to look at congruence and/or discordance between the objective of the study and the selected implementation scien

---

## [Decision Letter · Decision Letter 1]

20 Apr 2026

Use of Implementation Science Models, Theories, and Frameworks in Pediatric Rehabilitation: Protocol for a Scoping Review

PONE-D-25-39248R1

Dear Dr. Munce,

We’re pleased to inform you that your manuscript has been judged scientifically suitable for publication and will be formally accepted for publication once it meets all outstanding technical requirements.

Kind regards,

Udoka Okpalauwaekwe, MD, MPH, PhD

Academic Editor

PLOS One

Additional Editor Comments (optional):

I believe most of the reviewer comments have been addressed at this point hence wold move it forward for publication.

Reviewers' comments:

Reviewer's Responses to Questions

Comments to the Author

1. Does the manuscript provide a valid rationale for the proposed study, with clearly identified and justified research questions?

Reviewer #2: Yes

2. Is the protocol technically sound and planned in a manner that will lead to a meaningful outcome and allow testing the stated hypotheses?

Reviewer #2: Yes

3. Is the methodology feasible and described in sufficient detail to allow the work to be replicable?

Reviewer #2: Yes

4. Have the authors described where all data underlying the findings will be made available when the study is complete?

Reviewer #2: Yes

5. Is the manuscript presented in an intelligible fashion and written in standard English?

Reviewer #2: Yes

6. Review Comments to the Author

You may also provide optional suggestions and comments to authors that they might find helpful in planning their study.

Reviewer #2: Thanks for you attention to reiwer comments. I have no further comments at this time and look forward to reading from you in the future.

7. PLOS authors have the option to publish the peer review history of their article (what does this mean?). If published, this will include your full peer review and any attached files.

Do you want your identity to be public for this peer review? For information about this choice, including consent withdrawal, please see our Privacy Policy.

Reviewer #2:  Yes: Udoka Okpalauwaekwe

---

## [Editor Report · Acceptance letter]

PONE-D-25-39248R1

PLOS One

Dear Dr. Munce,

I'm pleased to inform you that your manuscript has been deemed suitable for publication in PLOS One. Congratulations! Your manuscript is now being handed over to our production team.

Kind regards,

on behalf of

Dr. Udoka Okpalauwaekwe

Academic Editor

PLOS One